# FAST PHYSICS-INFORMED LEARNING VIA DIFFUSION HYPERNETWORKS

## ABSTRACT

Physics-Informed Neural Networks (PINNs) have emerged as a powerful tool for solving partial differential equations (PDEs), and they have become a key workhorse in many AI-for-science applications. However, PINNs remain highly sensitive to factors such as initial conditions, domain geometries, and physical parameters. As a result, they typically require full retraining when these PDE-defining parameters change. In this work, we propose a diffusion-based hypernetwork that distills knowledge from training data to substantially accelerate PINN training. Our approach leverages a denoising diffusion probabilistic framework to generate PINN weights conditioned on PDE parameters. Once trained, the hypernetwork can directly produce PINNs for a family of parametric PDEs without requiring additional optimization. For more complex problems, the generated weights, used as initializations, reduce the training time by approximately $46\%$ for the Burgers1D-complex dataset and $60\%$ for the Wave2D dataset. Furthermore, the model demonstrates robustness to out-of-distribution PDE parameters, extending its applicability beyond the training distribution.

## 1 INTRODUCTION

Artificial intelligence has rapidly become a transformative tool in scientific discovery, enabling breakthroughs in domains such as weather forecasting (Price et al., 2025), materials design (Moosavi et al., 2020), climate modeling (Nguyen et al., 2023), and biomedical simulations (Zhang et al., 2024). At the core of many of these advances is the need to efficiently and accurately solve partial differential equations (PDEs), which govern a broad range of physical and engineering systems. In domains that demand repeated PDE solutions, such as parametric studies, uncertainty quantification, and real-time control, this necessity motivates the development of scalable, data-driven approaches that can effectively leverage prior knowledge to solve the PDEs, even when their parameters change.

Physics-informed neural networks (PINNs) (Raissi et al., 2019; Cuomo et al., 2022; Rathore et al., 2024) have emerged as a particularly promising tool for solving PDEs. PINNs work by enforcing physical constraints through PDE residuals, initial conditions, and boundary conditions. Their mesh-free formulation allows for flexible handling of complex geometries, irregular domains, and sparse observations, making them suitable for a wide range of scientific problems. Moreover, their compatibility with automatic differentiation and modern deep learning frameworks provides a seamless interface for integrating data and physics within a unified computational model. This combination of flexibility and rigor has enabled applications in multiple topics (Rao et al., 2021; Lu et al., 2021b; Zhang et al., 2024). Consequently, PINNs have quickly become a cornerstone methodology in the broader effort to merge machine learning with scientific computing, bridging data-driven inference with traditional physics-based simulation.

Despite their success, PINNs suffer from several practical limitations (Rathore et al., 2024). A key challenge lies in their sensitivity to problem-specific factors, such as initial conditions, domain geometries, and physical parameters. Even small variations in these settings typically require retraining of a PINN, often from scratch, which is time-consuming and computationally intensive. This limitation severely hinders their scalability to parametric PDE families, where one must solve the same governing equation across many different configurations. A natural idea is to treat problems with varying PDE parameters as separate tasks and apply meta-learning to transfer knowledge from previous tasks to new ones. However, our experiments show that existing meta-learning methods (Finn

et al., 2017; Nichol et al., 2018) fail to provide any gain in speed or accuracy. We attribute this to the large scale of PDE problems we aim to solve and the highly nonlinear dependence of weights of PINNs on their corresponding PDE parameters, which define each problem instance.

To address this challenge, we propose a diffusion-based hypernetwork framework, termed **HD-PINN**[1] (Hyper Diffusion for PINNs), to accelerate PINN training. Our approach leverages denoising diffusion probabilistic models (Ho et al., 2020) to learn a distribution over PINN weights conditioned on PDE parameters such as physical coefficients (e.g., viscosity or diffusivity), initial and boundary conditions, and domain geometry. By training on data pairs of varying PDE parameters and their corresponding PINN weights that solve the associated problems, the hypernetwork learns to directly generate PINN solvers for a target family of parametric PDEs, eliminating the need for additional optimization. For more complex PDE classes, the hypernetwork generated weights provide a high-quality initialization, substantially reducing overall training time by approximately

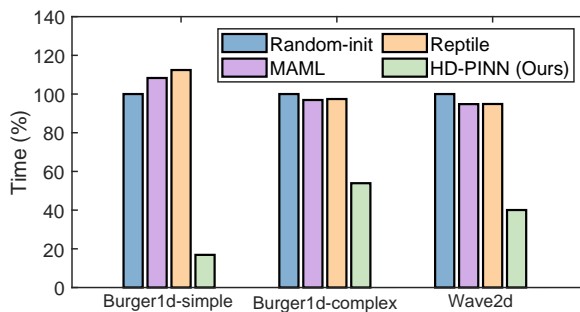

Figure 1: Percentage of training time using weights generated by different methods. The time for training from random initialization is used as the baseline (100%). Our HD-PINN significantly reduces training time compared to training from random initialization and meta-learning methods such as MAML (Finn et al., 2017) and Reptile (Nichol et al., 2018).

$46\%$ for the Burgers1D-complex dataset and $60\%$ for the Wave2D dataset, as noted in Figure 1. Moreover, our experiments show that the trained diffusion-based hypernetwork can provide high-quality initializations even for problems with PDE parameters outside the training distribution, as long as the deviations remain within a certain range.

Our main contributions are summarized as follows:

- **Diffusion-based hypernetwork for PINNs.** To the best of our knowledge, our method is the first diffusion-driven hypernetwork successfully applied to PINNs, capable of generating network weights conditioned on diverse PDE parameters;

- **Direct PINN weights generation.** Once trained, our model directly produces PINN solvers for a family of parametric PDEs without requiring additional optimization, effectively functioning as a mesh-free solver;

- **Accelerated PINN training via initialization.** For problems with higher variability, the generated weights provide high-quality initializations that substantially reduce training time while maintaining or improving solution accuracy;

- **Robustness to distribution shift.** The hypernetwork demonstrates robustness to out-of-distribution PDE parameters, extending its applicability beyond the training distribution.

## 2 RELATED WORK

### 2.1 AI FOR PHYSICS

Deep neural networks have emerged as a powerful tool for scientific computing, providing novel approaches to solve PDEs and model complex physical phenomena. Among these methods, Physics-Informed Neural Networks (PINNs) (Raissi et al., 2019) have received significant attention for their ability to embed the governing equations directly into the training objective. This approach allows PINNs to learn solutions to PDEs in a mesh-free manner while strictly enforcing physical constraints. Over the past few years, numerous extensions have been proposed to improve their practical performance, including strategies for faster convergence (Wang et al., 2021), adaptive loss weighting to balance competing objectives (Gao et al., 2025), scalable domain decomposition for large-scale problems (Shukla et al., 2021), and specialized designs for challenging scenarios

---

[1]Code is available at `anonymous.4open.science/r/HD-PINN-official-412B`.

such as multiphase flows and inverse problems (Sun et al., 2020). PINNs have shown success in a wide range of scientific domains, from fluid dynamics, including incompressible Navier–Stokes simulations (Raissi et al., 2020) and cardiovascular hemodynamics (Kissas et al., 2020), to solid mechanics for elasticity analysis (Rao et al., 2021).

Beyond PINNs, operator learning methods aim to generalize across families of PDEs by directly approximating mappings between function spaces. Representative examples include DeepONet (Lu et al., 2021a) with its branch–trunk design, and the Fourier Neural Operator (Li et al., 2020), which captures long-range dependencies efficiently in Fourier space. These approaches offer faster inference and improved generalization across parametric PDEs, but remain largely data-driven rather than physics-informed, making them sensitive to data quality and resolution (McCabe et al., 2023; Morel et al., 2025). Moreover, many operator-learning methods focus on short-term predictions and often struggle with stability when extrapolating far into the future.

Together, these developments highlight a growing trend toward data-driven, modular, and transfer-oriented architectures for accelerating PDE solvers. However, unlike our proposed framework, most prior methods remain limited to single-instance PDEs and/or involve substantial computational costs.

### 2.2 HYPERNETWORKS

Hypernetworks, introduced by Ha et al. (2016), are neural networks that generate the weights of a target network, allowing parameterization conditioned on specific inputs (Chauhan et al., 2024). This decouples task representations from task-specific training and enables flexible multi-task learning, conditional modeling, and rapid adaptation. Since their introduction, hypernetworks have been extended in diverse contexts. For example, Peebles et al. (2022) employ conditional diffusion transformers to generate weights based on prompted losses, while Erkoç et al. (2023) use unconditional diffusion models to produce MLP weights for neural implicit fields (Mildenhall et al., 2021), achieving efficient 3D/4D shape synthesis. These successes motivate our application of diffusion-driven hypernetworks to PINNs, a direction unexplored in AI for science. Other efforts pursue efficient weight generation and compression: Hedlin et al. (2025) compress full training trajectories into single-step estimates, Morel et al. (2025) generate operator-network parameters from short PDE trajectories for fast integration, and Cho et al. (2023) propose Hyper-LR-PINN, which reduces complexity by producing low-rank PINN representations. Unlike our framework, Hyper-LR-PINN is restricted to scalar inputs and cannot handle field-level variations such as initial conditions, domains, or spatially varying coefficients.

### 2.3 META-LEARNING

Our work is also closely related to meta-learning approaches (Finn et al., 2017; Nichol et al., 2018), which aim to reduce training costs by transferring knowledge across tasks. However, in parametric PDE families with high variability, shared initializations often fail to generalize and require extensive fine-tuning, as evidenced by our experiments and already illustrated in Figure 1. In contrast, our method learns a conditional mapping from PDE parameters to solver weights, providing task-specific initialization and overcoming the scalability limits of conventional meta-learning.

## 3 METHODS

### 3.1 PRELIMINARIES

**Physics-Informed Neural Network.** We consider a general partial differential equation (PDE) defined on a spatial-temporal domain:

$$
\begin{aligned}
\mathcal{F}[\boldsymbol{u}](\boldsymbol{x}, t) &= 0, & \boldsymbol{x} \in \Omega, \ t \in (0, T], \\
\mathcal{B}[\boldsymbol{u}](\boldsymbol{x}, t) &= 0, & \boldsymbol{x} \in \partial\Omega, \ t \in (0, T], \\
\boldsymbol{u}(\boldsymbol{x}, 0) &= \boldsymbol{u}_0(\boldsymbol{x}), & \boldsymbol{x} \in \Omega,
\end{aligned}
\tag{1}
$$

where $\boldsymbol{u}(\boldsymbol{x}, t)$ is the solution of interest, $\mathcal{F}[\cdot]$ is a differential operator, $\mathcal{B}[\cdot]$ denote the boundary condition operator, $\boldsymbol{u}_0(\boldsymbol{x})$ is the initial condition, $\Omega$ is the spatial domain and $[0, T]$ is the time interval.

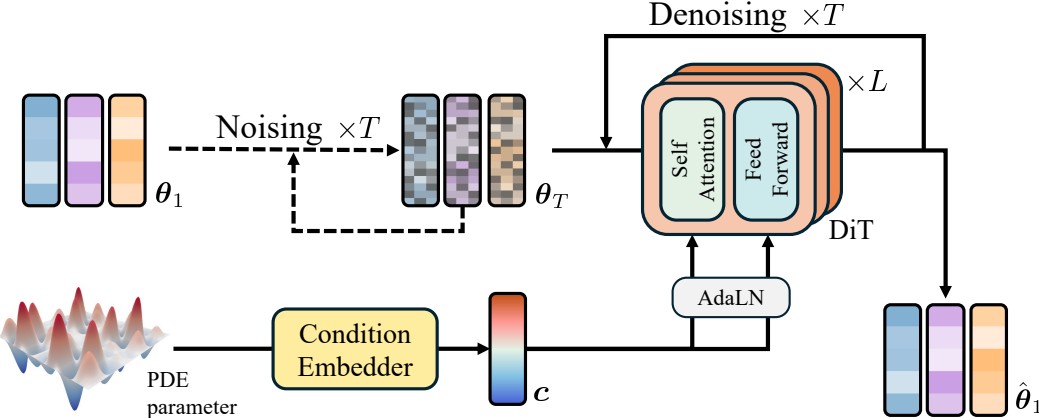

Figure 2: Illustration of the proposed HD-PINN framework. During training, PINN weights $\boldsymbol{\theta}_1$ are progressively noised into $\boldsymbol{\theta}_T$. A DiT module is then trained to reverse this process, denoising and reconstructing PINN weights $\hat{\boldsymbol{\theta}}_1$ conditioned on $\boldsymbol{c}$, which encodes the PDE parameters.

To train a PINN model $\boldsymbol{u}_{\boldsymbol{\theta}}$ parameterized by weights $\boldsymbol{\theta} \in \mathbb{R}^d$, which allows computation of $\boldsymbol{u}(\boldsymbol{x}, t) = \boldsymbol{u}_{\boldsymbol{\theta}}(\boldsymbol{x}, t)$, a set of collocation points is first generated, including $N_f$ points $(x_i^f, t_i^f)$ uniformly sampled in the whole domain $\Omega \times (0, T]$, $N_b$ points $(x_i^b, t_i^b)$ uniformly sampled on the boundary $\partial\Omega \times (0, T]$, and $N_0$ points $(x_i^0, t_i^0)$ uniformly sampled at the initial time domain $\Omega \times \{0\}$. When observational or synthetic data are available, additional $N_d$ points $(x_i^d, t_i^d)$ sampled from available data are incorporated to further constrain the solution. At each of these points, the neural network is evaluated and automatic differentiation is used to compute the residuals of the governing PDE, initial conditions, and boundary conditions. With the evaluated variable values and differentiations, the total loss $\mathcal{L}$ is written as a composition of multiple terms with weights that balance the contributions of each loss term,

$$\mathcal{L}_{\text{pinn}} = \lambda_{\text{data}}\mathcal{L}_{\text{data}} + \lambda_{\text{pde}}\mathcal{L}_{\text{pde}} + \lambda_{\text{bc}}\mathcal{L}_{\text{bc}} + \lambda_{\text{ic}}\mathcal{L}_{\text{ic}}, \tag{2}$$

where

$$\mathcal{L}_{\text{pde}} = \frac{1}{N_f} \sum_{i=1}^{N_f} \left\| \mathcal{F}[\boldsymbol{u}_{\boldsymbol{\theta}}](x_i^f, t_i^f) \right\|^2, \qquad \mathcal{L}_{\text{bc}} = \frac{1}{N_b} \sum_{i=1}^{N_b} \left\| \mathcal{B}[\boldsymbol{u}_{\boldsymbol{\theta}}](x_i^b, t_i^b) \right\|^2,$$

$$\mathcal{L}_{\text{data}} = \frac{1}{N_d} \sum_{i=1}^{N_d} \left\| \boldsymbol{u}_{\boldsymbol{\theta}}(x_i^d, t_i^d) - \boldsymbol{u}(x_i^d, t_i^d) \right\|^2, \qquad \mathcal{L}_{\text{ic}} = \frac{1}{N_0} \sum_{i=1}^{N_0} \left\| \boldsymbol{u}_{\boldsymbol{\theta}}(x_i^0, 0) - \boldsymbol{u}_0(x_i^0) \right\|^2. \tag{3}$$

**Regularized Weight Space Distribution.** Neural networks like PINNs often admit multiple local minima that yield comparable solutions, which can lead to a scattered and unstructured weight space. To facilitate the construction of a well-behaved dataset of optimized weights and support downstream hypernetwork training, we adopt two complementary strategies. First, all PINNs are initialized from a shared set of pre-trained weights $\boldsymbol{\theta}_{\text{init}}$, ensuring a consistent starting point across training instances (Erkoç et al., 2023). Second, during the collection of optimized weights, we add a distance regularization term that penalizes deviations from $\boldsymbol{\theta}_{\text{init}}$. This encourages a more compact and coherent weight space and prevents solutions from drifting too far from a common reference. Mathematically, the regularized loss is written as

$$\tilde{\mathcal{L}}_{\text{pinn}} = \mathcal{L}_{\text{pinn}} + \lambda_{\text{reg}} \|\boldsymbol{\theta} - \boldsymbol{\theta}_{\text{init}}\|_2, \tag{4}$$

where $\lambda_{\text{reg}}$ controls the strength of the regularization. This approach produces a structured weight landscape, which not only improves stability during training but also facilitates learning of a hypernetwork that can generate effective weights for a variety of PDE instances.

## 3.2 Conditional Weight-Space Diffusion

**Problem Setting.**   Figure 2 illustrates the proposed HD-PINN architecture. The central objective is to learn a mapping from task-specific PDE conditions to a structured distribution of PINN weights, effectively capturing the relationship between PDE parameters and corresponding network solutions. As formalized in Equation 1, the PDE conditions can include physical coefficients (e.g., viscosity, diffusivity, or density), initial states, boundary conditions, and domain representations such as binary masks indicating spatial support. To encode these diverse inputs, we employ a conditioner encoder that produces a compact conditioning vector $c$. Denote $\theta_0 \sim p_{\text{data}}(\theta \mid c)$ as the optimized PINN weights for a given task. The forward diffusion process then progressively perturbs these weights according to

$$\theta_t = \gamma(t)\,\theta_0 + \eta(t)\,\epsilon, \qquad \epsilon \sim \mathcal{N}(\mathbf{0}, \mathbf{I}),$$

where $\gamma(t)$ and $\eta(t)$ define the noise schedule. The model is trained to learn a *conditional score function*

$$s_\phi(\theta, t, c) \approx \nabla_\theta \log p_t(\theta \mid c), \tag{5}$$

which estimates the gradient of the log-density of the noised weight distribution at time $t$, conditioned on the PDE parameters (Vincent, 2011; Song & Ermon, 2019; Song et al., 2021).

At inference time, ground-truth PINN weights are no longer available, so the operation indicated by the dotted line in Figure 2 is omitted. For a new task with conditions $c_{\text{new}}$, the reverse-time stochastic differential equation (or equivalently the probability-flow ODE) is integrated, guided by the learned conditional score function $s_\phi$. This reverse diffusion process transforms Gaussian noise into a structured set of weights $\hat{\theta}_0(c_{\text{new}})$ that can directly produce or initialize a PINN tailored to the new PDE instance. This enables rapid deployment of PINN solvers across a family of parametric PDEs without retraining from scratch.

**Training Objective.**   To optimize $\phi$, we adopt the denoising score matching (DSM) loss from DDPM (Ho et al., 2020). The network $\epsilon_\phi$ is trained to recover Gaussian noise added to the target PINN parameters. The training loss is defined as

$$\mathcal{L}_{\text{DSM}} = \mathbb{E}_{(c,\theta)\sim\mathcal{D}} \, \mathbb{E}_{\tau\sim\mathcal{U}(1,T)} \, \mathbb{E}_{\epsilon\sim\mathcal{N}(0,I)} \left[ \left\| \epsilon - \epsilon_\phi \left( \sqrt{\bar{\gamma}_\tau}\,\theta + \sqrt{1 - \bar{\gamma}_\tau}\,\epsilon, \, \tau, \, c \right) \right\|_2^2 \right], \tag{6}$$

where $\mathcal{D}$ denotes the data distribution, and $\bar{\gamma}_\tau$ denotes the cumulative product of noise schedule coefficients. The model receives as input a noisy version of the target PINN parameters, the diffusion step $\tau$, and the PDE conditions $c$, and learns to denoise and recover the original PINN weights $\theta$.

**Condition Dropout.**   To further enhance robustness and prevent overfitting, we apply dropout to the conditioning vector $c$ during training. This acts as a form of implicit ensemble regularization (Srivastava et al., 2014), discouraging co-adaptation among the conditioning inputs and promoting smoother mappings from PDE parameters to generated weights. The use of condition dropout helps stabilize the diffusion process, improves generalization to unseen tasks, and reduces the sensitivity of the hypernetwork to small perturbations in the input conditions.

## 4 Experimental Results

We evaluated our HD-PINN framework on multiple standard benchmark datasets that span different PDE types and complexities. Our experiments demonstrate the effectiveness of hypernetwork-generated initializations to accelerate PINN training while maintaining the solution accuracy highlighted in Figure 1.

### 4.1 Implementation Details

**Dataset.**   We consider three datasets. `Burgers1D-simple` consists of 1D Burgers' equations with varying initial conditions $u_0(x) = \alpha_1(x-1)(x+1)(x+\alpha_2)$, where $\alpha_1$ and $\alpha_2$ are uniformly sampled from $(0.5, 1.5)$ and $(-0.5, 0.5)$, respectively. `Burgers1D-complex` is derived from PDEBench (Takamoto et al., 2022) with a broader set of varying initial conditions. `Wave2D` is constructed by varying irregular PDE domains specified through binary masks. Additional details are provided in Appendix A, while the details for generating random PDE domains for `Wave2D` are given in Appendix B.

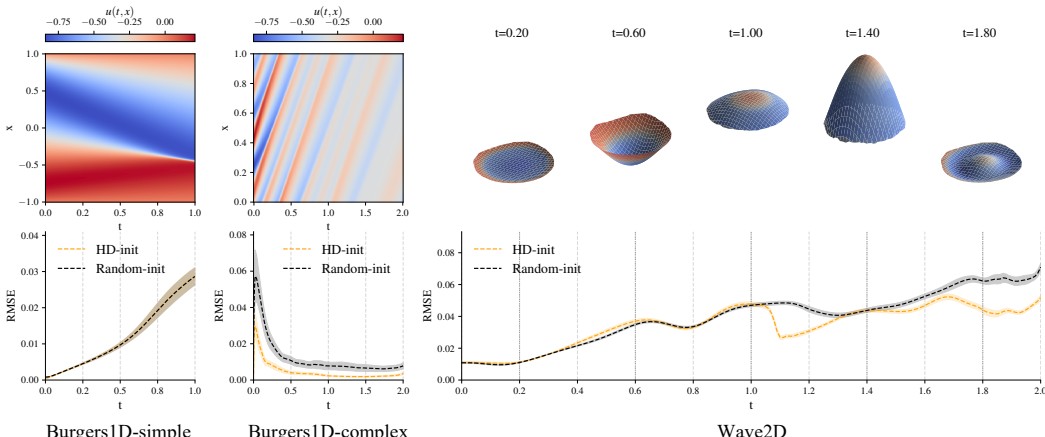

Figure 3: Evolution of representative solution fields and the associated RMSE statistics, where dotted lines indicate the mean and shaded regions denote standard deviations, across different time steps for the PDEs: (left) `Burgers1D-simple` tasks, PINNs trained completely unsupervised; (middle) `Burgers1D-complex` dataset with more challenging initial conditions, PINNs are trained weakly supervised using $0.5\%$ ground-truth data; (right) `Wave2D` dataset, PINNs are trained weakly supervised using $3\%$ ground-truth data. **HD-init** is capable of generating high-quality initializations that lead to accurate solutions with low error across different time steps.

**Comparing Methods.**  In addition to closely related meta-learning methods such as **MAML** (Finn et al., 2017) and **Reptile** (Nichol et al., 2018), we compare against some baselines. The first is **Random-init**, where each new PINN is trained from scratch with randomly initialized weights. For our proposed approach, we consider **HD-direct**, which denotes directly using the PINN weights generated by the hypernetwork without refinement, and **HD-init**, which uses these generated weights as initialization followed by fine-tuning.

**Model and Training.**  PINNs are implemented as small MLPs trained with Adam followed by L-BFGS. The hypernetwork is based on a DiT backbone with task-specific encoders: an MLP for `Burgers1D` initial conditions and a CNN for `Wave2D` domain masks. The module architectures, conditioner encoders, and training details are provided in Appendix C, D, and E, respectively.

**Metrics.**  We evaluated all methods by wall-clock training time and accuracy, measured using root mean square error (RMSE) and normalized relative $L^2$ error (Rel. Error), with reference solutions obtained via finite differences (Table 1). For ablation, we used the distance correlation (dCor) (Székely et al., 2007) to quantify the dependence between the PDE parameters and the optimized PINN weights; the definition is provided in Appendix F and the results are in Appendix G.

## 4.2 PERFORMANCE ON IN-DISTRIBUTION TEST DATA

### 4.2.1 DIRECT HYPERNETWORK PREDICTIONS.

We first evaluate our method on the `Burgers1D-simple` dataset to demonstrate the ability of HD-PINN to directly generate PINN solvers without additional fine-tuning. In this setup, the hypernetwork takes the initial condition as input and outputs PINN weights that are immediately applied to solve the 1D Burgers' equation, yielding ready-to-use solvers without iterative training.

Figure 3 illustrates a representative trajectory and error evolution, while Table 1 shows that HD-direct achieves accuracy comparable to the Random-init baseline, confirming that the generated weights yield reasonable solutions. For completeness, we also report HD-init, where hypernetwork-generated weights are fine-tuned, requiring minimal extra training. Unlike meta-learning approaches such as MAML (Finn et al., 2017) and Reptile (Nichol et al., 2018), which rely on a single shared initialization and struggle with diverse PDE families, HD-PINN explicitly learns a task-to-weights mapping, enabling task-specific solvers.

| Dataset | Method | Time (s) ↓ | RMSE ($10^{-2}$) ↓ | Rel. Error (%) ↓ |
|---|---|---|---|---|
| Burgers1D-simple | Random-init | 17.5 ± 7.2 | **1.6 ± 0.8** | 4.7 ± 1.2 |
| | MAML (Finn et al., 2017) | 18.9 ± 6.8 | **1.6 ± 0.8** | **4.6 ± 1.3** |
| | Reptile (Nichol et al., 2018) | 19.6 ± 7.1 | **1.6 ± 0.8** | **4.6 ± 1.3** |
| | **HD-direct (ours)** | - | 2.8 ± 1.3 | 9.0 ± 4.1 |
| | **HD-init (ours)** | **3.0 ± 3.9** | **1.6 ± 0.8** | 4.7 ± 1.2 |
| Burgers1D-complex | Random-init | 180.9 ± 28.7 | 1.1 ± 1.6 | 2.5 ± 4.7 |
| | MAML (Finn et al., 2017) | 175.3 ± 16.2 | 1.0 ± 1.5 | 2.2 ± 3.6 |
| | Reptile (Nichol et al., 2018) | 176.2 ± 16.4 | 1.0 ± 1.5 | 2.2 ± 3.8 |
| | **HD-direct (ours)** | - | 45.2 ± 32.7 | 109.9 ± 118.7 |
| | **HD-init (ours)** | **97.5 ± 34.6** | **0.8 ± 1.0** | **1.7 ± 2.9** |
| Wave2D | Random-init | 519.3 ± 59.4 | 3.8 ± 0.4 | 4.2 ± 0.4 |
| | MAML (Finn et al., 2017) | 509.9 ± 64.4 | 3.8 ± 0.3 | 4.1 ± 0.3 |
| | Reptile (Nichol et al., 2018) | 510.3 ± 65.0 | 3.8 ± 0.3 | 4.1 ± 0.3 |
| | **HD-direct (ours)** | - | 16.6 ± 15.8 | 17.7 ± 16.7 |
| | **HD-init (ours)** | **208.5 ± 113.3** | **3.6 ± 0.6** | **3.9 ± 0.6** |

Table 1: Performance comparison for different PINN modes among **Random-init**, **HD-direct** (directly using the weights generated without refinement), and **HD-init** (uses these generated weights as initialization followed by fine-tuning) on `Burgers1D` and `Wave2D` after fine-tuning. For `Burgers1D-simple`, HD-PINN directly yields accurate solutions with negligible need for tuning. For more challenging cases, the generated PINN weights reduce training time by approximately 46% on `Burgers1D-complex` and 60% on `Wave2D`. Bold indicates the best. Shadow indicates the second.

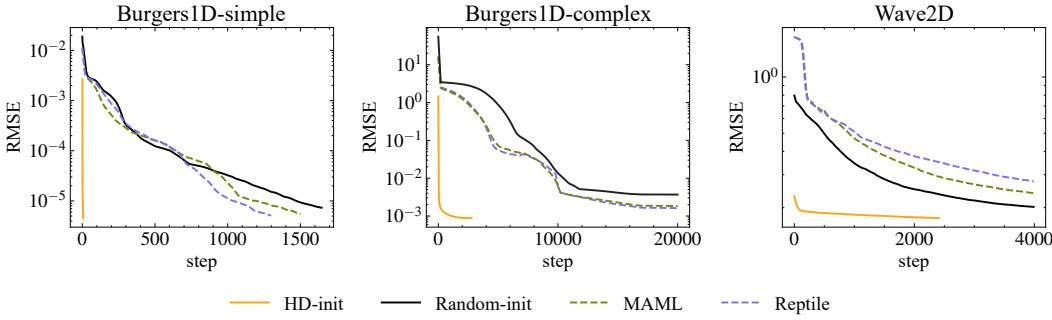

Figure 4: Training error plots comparing **Random-init**, **MAML**, **Reptile**, and our **HD-init** on `Burgers1D-simple`, `Burgers1D-complex` and `Wave2D`. The step count denotes L-BFGS iterations until stopping. Additional examples can be found in Appendix I.

### 4.2.2 REFINE FROM HYPERNETWORK INITIALIZATION.

On simple PDEs, **HD-direct** can generate near-optimal PINN weights. For more complex PDEs, however, the ill-conditioned loss landscape makes direct weight generation unreliable, as small parameter perturbations can cause large degradation (Rathore et al., 2024). In such cases, **HD-init** serves as a prior-informed initialization: diffusion-conditioned weights that remain near-optimal and greatly reduce training time. To test generality, we evaluate on more challenging datasets, `Burgers1D-complex` and `Wave2D`, which involve substantial parameter variations and hinder direct weight generation. Even so, our method consistently yields high-quality initializations that accelerate convergence; fine-tuning details are provided in Appendix E.

Quantitative comparisons, summarized in Table 1, show that PINNs initialized with HD-init consistently attain high-performance solutions in much shorter training times compared to Random-init, while maintaining or even slightly improving accuracy. This indicates that HD-PINN successfully captures generalized structure across effective PINN weights for different PDE instances, enabling the generation of problem-specific initializations that accelerate training. In contrast, conventional

| OOD shift | Method | Time (s) ↓ | Rel. Error (%) ↓ |
|---|---|---|---|
| No shift | Random-init | $17.5 \pm 7.2$ | **$4.7 \pm 1.2$** |
|  | **HD-init** | **$3.0 \pm 3.9$** | **$4.7 \pm 1.2$** |
| $\delta = 0.1$ | Random-init | $15.9 \pm 10.9$ | **$4.0 \pm 1.6$** |
|  | **HD-init** | **$5.9 \pm 8.7$** | **$4.0 \pm 1.6$** |
| $\delta = 0.2$ | Random-init | $23.3 \pm 15.9$ | **$4.9 \pm 1.9$** |
|  | **HD-init** | **$13.7 \pm 13.3$** | **$4.9 \pm 1.9$** |
| $\delta = 0.3$ | Random-init | $24.7 \pm 19.4$ | $5.1 \pm 1.8$ |
|  | **HD-init** | **$15.5 \pm 14.1$** | **$5.0 \pm 1.9$** |
| $\delta = 0.4$ | Random-init | $23.9 \pm 22.6$ | **$5.2 \pm 1.9$** |
|  | **HD-init** | **$18.5 \pm 17.6$** | $5.3 \pm 2.4$ |

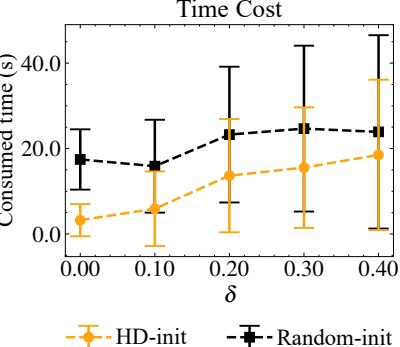

Table 2: Out-of-distribution (OOD) evaluation on the `Burgers1D-simple` dataset with varying initial conditions (table, left) and corresponding time-to-convergence plot (right). The hypernetwork is trained on $\alpha_1 \sim \mathcal{U}(0.5, 1.5)$ and $\alpha_2 \sim \mathcal{U}(-0.5, 0.5)$, and tested under shifted regimes of $\alpha_1$. The shift $\delta$ offsets the training interval, yielding the evaluation domain $(0.5 - \delta, 0.6 - \delta) \cup (1.4 + \delta, 1.5 + \delta)$. No shift corresponds to the original training range. Bold indicates the best result at the shift level.

meta-learning approaches such as MAML and Reptile, which rely on a shared initialization across tasks, provide only marginal speed-ups, likely due to the high diversity and large number of tasks involved in these datasets.

To provide a more intuitive understanding, we visualize representative solutions and corresponding error distributions over time for `Burgers1D-complex` and `Wave2D` in the second and third columns of Figure 3. These visualizations highlight that PINNs initialized with HD-PINN maintain low errors across all PDEs time steps, demonstrating the robustness and accuracy of the generated weights.

Additionally, training logs for representative PINNs on both datasets are shown in Figure 4. The plots demonstrate that PINNs initialized with HD-init converge more rapidly and achieve high-accuracy solutions earlier than Random-init. In comparison, MAML and Reptile show modest improvements in only some cases, with additional examples provided in Appendix I. These observations highlight that HD-PINN generates high-quality, task-specific initializations that substantially accelerate training, even under significant parameter variability. Overall, the results confirm that our framework effectively captures transferable knowledge across diverse PDE instances, enabling the rapid deployment of PINN solvers for more complex problems.

### 4.3 OUT-OF-DISTRIBUTION

To rigorously evaluate the out-of-distribution (OOD) robustness of our framework, we asked whether HD-PINN can generalize beyond the training range of PDE parameters and remain useful under unseen regimes. This is a crucial step in assessing whether the method can scale to scientific problems where parameters are not confined to a pre-defined distribution.

#### 4.3.1 OOD FOR BURGERS1D-SIMPLE.

In this test, we trained the hypernetwork on the `Burgers1D-simple` dataset with initial conditions parameterized by $\alpha_1 \sim \mathcal{U}(0.5, 1.5)$. We then systematically shifted $\alpha_1$ outside this range to create OOD test sets. The shift is defined relative to the boundaries of the training distribution: for a shift $\delta$, the new evaluation domain becomes $(0.5 - \delta, 0.6 - \delta) \cup (1.4 + \delta, 1.5 + \delta)$. This setup allows us to control the degree of OOD difficulty. For example, $\delta = 0.1$ corresponds to a mild extrapolation $(0.4, 0.5) \cup (1.5, 1.6)$, while larger values such as $\delta = 0.4$ represent more severe shifts.

The results in Table 2 show that hypernetwork initialization (HD-init) consistently accelerates convergence compared to random initialization (Random-init), even under parameter shifts. This indicates that the hypernetwork encodes transferable structural information about the PDE family. For small shifts, HD-init yields stable error reduction and significant time savings, while for larger shifts the benefits gradually diminish, as also seen in the associated figure. Overall, within a reasonable

| Diameter | Method | Time (s) ↓ | Rel. Error (%) ↓ |
|---|---|---|---|
| $l \approx 1.20$ | Random-init | $519.3 \pm 59.4$ | $4.1 \pm 0.3$ |
| | **HD-init** | $\mathbf{208.5 \pm 113.3}$ | $\mathbf{3.9 \pm 0.5}$ |
| $l \approx 1.25$ | Random-init | $521.0 \pm 58.0$ | $4.1 \pm 0.4$ |
| | **HD-init** | $\mathbf{256.8 \pm 128.5}$ | $\mathbf{3.9 \pm 0.6}$ |
| $l \approx 1.30$ | Random-init | $512.5 \pm 62.9$ | $4.1 \pm 0.4$ |
| | **HD-init** | $\mathbf{269.0 \pm 134.1}$ | $\mathbf{3.8 \pm 0.4}$ |
| $l \approx 1.35$ | Random-init | $517.6 \pm 62.0$ | $4.0 \pm 0.3$ |
| | **HD-init** | $\mathbf{305.4 \pm 120.1}$ | $\mathbf{3.8 \pm 0.7}$ |
| $l \approx 1.40$ | Random-init | $513.0 \pm 63.0$ | $4.0 \pm 0.4$ |
| | **HD-init** | $\mathbf{311.9 \pm 120.4}$ | $\mathbf{3.8 \pm 0.6}$ |

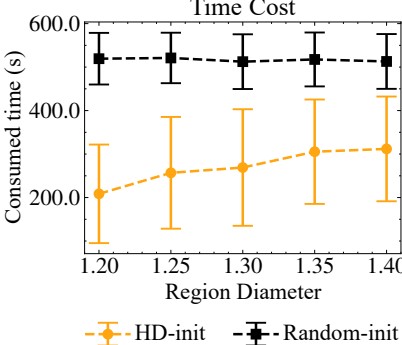

Table 3: Out-of-distribution (OOD) evaluation on the `Wave2D` dataset with varying domain diameters (table, left) and corresponding time-to-convergence plot (right). The hypernetwork is trained on domains with diameter $l \approx 1.20$ and tested on larger domains with diameters ranging from $l \approx 1.25$ to $l \approx 1.40$. Bold indicates the best result at the shift level.

extrapolation range, the hypernetwork provides effective priors that speed up optimization while remaining robust. For completeness, we also report the performance of HD-direct in Appendix H, where the hypernetwork's predicted weights are used without further optimization.

### 4.3.2 OOD FOR WAVE2D

For this experiment, we evaluate OOD performance using a hypernetwork trained on the `Wave2D` dataset from Section 4.2, where the PDE parameter is the domain diameter $l \approx 1.20$. By modifying the parameters of the mask generation algorithm described in Appendix B, we create four OOD test sets with diameters $l \approx 1.25, 1.30, 1.35$, and $1.40$. The generated masks are then used as inputs to the hypernetwork, which produces initialization weights for solving the 2D wave equation on each corresponding domain.

The results for the Wave2D OOD test are presented in Table 3. Similar to the Burgers1D-simple case, we observe that using hypernetwork-generated weights as initialization (HD-init) consistently accelerates convergence compared to random initialization (Random-init). This advantage persists across all tested domain diameters, indicating that the hypernetwork captures transferable structural information about PDE solutions that remains useful even when the evaluation domains deviate from the training regime. As the domain diameter increases, however, the benefits gradually diminish as shown in the associated plot, reflecting the greater difficulty of extrapolating to larger geometric shifts.

## 5 CONCLUSIONS

We introduced a diffusion-based hypernetwork for accelerating the training of physics-informed neural networks (PINNs) by generating their weights conditioned on problem-specific PDE parameters. Our approach distills knowledge on the relationship between the PDE parameters that define a problem and the weights of its corresponding PINN solver. Once trained, the hypernetwork supports two complementary modes: (*i*) direct inference, producing PINN weights that solve simple PDE instances without additional optimization; and (*ii*) high-quality initialization, providing substantial reductions in training cost for more complex problems. Moreover, for moderate out-of-distribution PDE parameters, the trained hypernetwork can still generate usable weights that accelerate PINN fine-tuning. Collectively, these capabilities point toward a promising pathway for fast, flexible, and data-driven PDE solvers, bridging generative modeling with scientific machine learning.

In future work, we plan to extend this work to a broader range of PDE-driven applications. We also aim to further improve the model architecture and training strategies to enhance scalability, generalization, and robustness.

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

## A  DATASETS

**Burgers1D-simple.**  The 1D Burgers' Equation is a widely studied nonlinear PDE that serves as a prototypical model for transport phenomena, combining the effects of nonlinear advection and diffusion. It is frequently used in numerical analysis and scientific machine learning as a benchmark due to its relative simplicity while still capturing essential features of more complex fluid dynamics systems. The equation is given by

$$\frac{\partial u}{\partial t} + u\frac{\partial u}{\partial x} = \frac{\nu}{\pi}\frac{\partial^2 u}{\partial x^2}, \tag{7}$$

where $u(x,t)$ denotes the scalar field of interest, $x$ is the spatial coordinate, $t$ is time, and $\nu$ is the viscosity coefficient controlling the strength of diffusion relative to nonlinear transport.

For `Burgers1D-simple` dataset, the problem is defined on the domain $x \in [-1, 1]$, $t \in [0, 1]$ with fixed viscosity $\nu = 0.01$ and homogeneous Dirichlet boundary conditions. The initial conditions are sampled from a parametric family of cubic polynomials, $u_0(x) = \alpha_1(x-1)(x+1)(x+\alpha_2)$, where $\alpha_1 \sim \mathcal{U}(0.5, 1.5)$ and $\alpha_2 \sim \mathcal{U}(-0.5, 0.5)$, allowing for a diverse set of trajectories. The corresponding PINN weights $\boldsymbol{\theta}$ are optimized without relying on data-informed loss, ensuring that the network relies solely on PDE residual minimization. This dataset contains 9,000 training pairs $(\alpha, \boldsymbol{\theta})$ and 1,000 testing pairs.

**Burgers1D-complex.**  This dataset is based on the PDEBench benchmark (Takamoto et al., 2022). While it also from Burgers' Equation in 7, it is defined with $x \in [0, 1]$, $t \in [0, 2]$, viscosity $\nu = 0.01$, and periodic boundary conditions, which yield more complex solution dynamics. Here, each PDE instance is uniquely determined by its initial condition $u_0(x)$, which governs the trajectory evolution. For these cases, PINNs are optimized with data-informed losses to stabilize training and improve solution accuracy. The dataset provides 9,000 $(u_0, \boldsymbol{\theta})$ training pairs and 1,000 testing pairs. Together, these two datasets allow us to evaluate both direct solver generation and initialization under increasingly challenging settings.

**Wave2D.**  We also construct a dataset for the 2D Wave Equation, which models the propagation of waves in heterogeneous or irregular domains. The governing PDE is

$$\frac{\partial^2 u}{\partial t^2} - \mu^2 \nabla^2 u = 0, \tag{8}$$

where $u(x, y, t)$ is the wave field and $\mu = 1$ is the wave speed. To induce nontrivial dynamics, the boundary condition $u_b(x, y, t) = 1 - \cos(\pi t)$ is imposed, generating inward-propagating waves from the domain boundary.

To increase problem diversity, the computational domains $\Omega$ are randomly generated using stochastic region growing, producing irregular shapes that challenge the generalization ability of PINN solvers. For each domain, a data-informed PINN is trained to approximate the wave dynamics, providing a realistic set of weights that encode both PDE physics and domain-specific geometry. The resulting dataset contains 15,000 $(\Omega, \boldsymbol{\theta})$ training pairs and 4,000 testing pairs.

## B  IRREGULAR DOMAIN GENERATION

To generate irregular computational domains for the `Wave2d` dataset, we design a simple yet effective mask generation procedure. The method starts from an initial disk placed at the center of the domain and iteratively grows the region by randomly activating neighboring pixels. This ensures that the resulting mask remains connected while allowing for irregular and non-symmetric shapes. To improve robustness, we retain only the largest connected component, smooth the boundaries via contour filling, and finally apply Gaussian blurring followed by thresholding to obtain clean binary masks. The process produces a diverse collection of irregular shapes that can be used as PDE domains for training and evaluation.

---

**Algorithm 1** Random Irregular Mask Generation

---

**Require:** mask size $(h, w)$, maximum pixels $P$, disk radius $r$
**Ensure:** binary irregular mask $M$
 1: Initialize $M \leftarrow 0^{h \times w}$ and visited map $V \leftarrow \text{False}^{h \times w}$
 2: Place an initial disk of radius $r$ at the center; set $M = 1$, $V = \text{True}$ inside the disk
 3: Set pixel counter $c \leftarrow 1$ to record number of active pixels in $M$
 4: **while** $c < P$ **do**
 5:      Dilate $M$ with a 4-connected structuring element
 6:      Identify candidate pixels $C \leftarrow \{(i, j) : \text{dilated}(i, j) = 1, V(i, j) = \text{False}\}$
 7:      **if** $C = \emptyset$ **then**
 8:          **break**
 9:      **end if**
10:      Randomly select $(i, j) \in C$
11:      Set $M[i, j] \leftarrow 1$, $V[i, j] \leftarrow \text{True}$
12:      $c \leftarrow c + 1$
13: **end while**
14: Retain only the largest connected component of $M$
15: Extract contour of $M$ and redraw it as a filled mask
16: Apply Gaussian blur and thresholding to obtain final binary mask
17: **return** $M$

---

## C    Model Architecture and Training Details

**PINN Architecture.** The PINNs are implemented as fully connected multilayer perceptrons (MLPs). For the 1D Burgers' Equation we use a 3-layer MLP with 20 hidden units per layer, while for the Wave Equation we adopt a slightly larger 4-layer MLP with 30 units per layer to capture richer dynamics. The activation function is $\tanh$, which is commonly used in PINNs due to its smoothness and stability for PDE approximation. Training follows a two-stage procedure: Adam (Kingma & Ba, 2015) with learning rate $1 \times 10^{-3}$ for initial exploration, followed by L-BFGS (Byrd et al., 1995) for fine-tuning. A weight distance regularization term with coefficient $\lambda_{\text{reg}} = 1 \times 10^{-5}$ is applied across all datasets to stabilize optimization.

**Hypernetwork Architecture.** The hypernetwork builds on the Diffusion Transformer (DiT) (Peebles & Xie, 2023), with customized linear layers to encode and decode the parameters of each PINN layer. We use a 12-layer transformer with 16 attention heads and hidden dimension 1,536. Task conditions $\boldsymbol{c}$ are injected through adaptive normalization (Perez et al., 2018) at every layer, enabling precise task-specific weight generation. To enable classifier-free guidance, we apply *token drop* with probability 0.1 during training, replacing dropped embeddings with a constant unconditional vector. At inference, embeddings are used deterministically.

The diffusion process is trained with a denoising score-matching loss under a linear noise schedule. Training is performed on a single NVIDIA A100 GPU with batch size 256. Once trained, HD-PINN can generate weights for many distinct PDE tasks within seconds, with inference time depending mainly on the number of sampling steps.

## D    Task-Specific Encoder

**Burgers1D.** For 1D Burgers' problems, the conditioning information comes from the sampled initial condition curve, which encodes the trajectory of the PDE solution. We represent this input using a 3-layer multilayer perceptron (MLP) with dimensions $[d, D, D, D]$, where $d$ is the number of sampled spatial points on the curve (default $d = 1,024$), and $D$ is the hidden size of the DiT backbone. Each hidden layer uses the GELU activation, chosen for its smooth nonlinearity and stability in high-dimensional settings. This encoder effectively compresses the high-dimensional curve input into a compact representation, while preserving the essential variation in initial conditions. The final output is a $D$-dimensional embedding that serves as the conditioning vector for the HD-PINN.

**Wave2D.** For 2D wave problems, the conditioning input is more complex, as it must capture the geometry of irregular computational domains. To encode these domain shapes, we design a 3-layer convolutional neural network (CNN) followed by a linear projection. The inputs are binary masks of size $32 \times 32$, where each pixel indicates whether the point lies inside the computational domain. The CNN applies three $5 \times 5$ convolutional layers with channel sizes $(1 \to D/4 \to D/2 \to D)$, each followed by `GELU` activation for stable feature extraction. Between convolutional layers, we apply progressive `AdaptiveAvgPool2d` operations to reduce the resolution $(H \to H/2 \to H/4 \to 1)$ while retaining global structural information. The resulting feature map is globally pooled and flattened into a $D$-dimensional vector, which acts as the conditioning input to HD-PINN.

## E  FINE-TUNING FROM HD-INIT

We report the performance of PINN finetuned from HD-init on the test set from `Burgers1D-complex` and `Wave2D` dataset in Table 1. The finetuned PINNs are initialized with the weights generated from HD-init, and then finetuned with the L-BFGS optimizer until convergence. The L-BFGS optimizer is configured with a learning rate of 1.0, a maximum of 10,000 iterations, and a tolerance of $1 \times 10^{-9}$. The training stops when either the maximum iterations or the tolerance condition is met. For the other methods, the initialized PINNs are tuned using Adam followed by L-BFGS (Rathore et al., 2024; Cuomo et al., 2022), under the same configurations.

## F  DISTANCE-CORRELATION

To quantify the statistical dependence between two random variables $X \in \mathbb{R}^p$ and $Y \in \mathbb{R}^q$, we employ the distance correlation (dCor) (Székely et al., 2007). Unlike the classical Pearson correlation, which only captures linear dependence, dCor detects both linear and nonlinear associations between distributions.

Given $n$ paired samples $\{(x_i, y_i)\}_{i=1}^n$, let

$$a_{ij} = \|x_i - x_j\|, \qquad b_{ij} = \|y_i - y_j\| \tag{9}$$

denote the pairwise Euclidean distance matrices. These are then double-centered to remove mean effects,

$$A_{ij} = a_{ij} - \bar{a}_{i\cdot} - \bar{a}_{\cdot j} + \bar{a}_{\cdot\cdot}, \quad B_{ij} = b_{ij} - \bar{b}_{i\cdot} - \bar{b}_{\cdot j} + \bar{b}_{\cdot\cdot}, \tag{10}$$

where $\bar{a}_{i\cdot}$ is the $i$th row mean of $a_{ij}$, $\bar{a}_{\cdot j}$ is the $j$th column mean, and $\bar{a}_{\cdot\cdot}$ is the grand mean (similarly for $b$).

The distance covariance is then defined as

$$\text{dCov}^2(X, Y) = \frac{1}{n^2} \sum_{i,j=1}^n A_{ij} B_{ij}, \tag{11}$$

and the corresponding distance correlation is

$$\text{dCor}(X, Y) = \frac{\text{dCov}(X, Y)}{\sqrt{\text{dCov}(X, X)\, \text{dCov}(Y, Y)}}. \tag{12}$$

By construction, $\text{dCor}(X, Y) \in [0, 1]$, where $\text{dCor}(X, Y) = 0$ if and only if $X$ and $Y$ are independent, and larger values indicate stronger dependence between the two distributions.

## G  CONTROLLED INITIALIZATION AND WEIGHT REGULARIZATION

We investigate the effect of the proposed weight-space regularization using the `Burgers1D-simple` dataset, with the aim of determining whether explicitly encouraging structure in the weight space improves the relationship between problem parameters and optimized PINN solutions. To this end, we measure the dependency between PDE parameters (here, the initial conditions) and the corresponding trained PINN weights using distance correlation (dCor) (Székely et al., 2007), which captures both linear and nonlinear associations. In addition, we report the RMSE

of PINN predictions relative to ground-truth trajectories computed with a finite-difference method (FDM) solver.

The results in Table 4 reveal two main observations. First, controlled initialization markedly increases the correlation between initial conditions and optimized weights, as reflected by the higher dCor values. Second, when the regularization strength $\lambda_{\text{reg}}$ is properly chosen (e.g., $10^{-5}$), the predictive accuracy does not deteriorate and even shows slight improvement compared to random initialization and unregularized training. This suggests that weight-space smoothing not only preserves accuracy but may also aid optimization by guiding the network toward more structured solutions.

Overall, these findings indicate that controlled initialization both stabilizes training and produces weight representations that are better aligned with PDE parameters, making them more suitable for downstream hypernetwork learning.

| | Random-init. | Controlled-init. | | | |
|---|---|---|---|---|---|
| | - | $\lambda_{\text{reg}} = 0$ | $\lambda_{\text{reg}} = 10^{-5}$ | $\lambda_{\text{reg}} = 10^{-4}$ | $\lambda_{\text{reg}} = 10^{-3}$ |
| dCor $\uparrow$ | 0.1313 | 0.5969 | 0.5972 | 0.5977 | **0.5994** |
| mean RMSE($10^{-3}$) $\downarrow$ | 4.8299 | 4.6288 | **4.5780** | 4.6538 | 4.6571 |

Table 4: Distance correlation (dCor) computed between 200 pairs of initial conditions and optimized weights from `Burgers1D-simple`. Bold indicates best result.

## H    DIRECT EVALUATION FOR OOD

To complement the OOD evaluation reported in Tables 2 and 3, we also present in Tables 5 and 6 the direct evaluation results on the `Burgers1D-simple` and `Wave2D` datasets. These results are included for completeness and should be regarded as supplementary to the main OOD analysis. As expected, accuracy decreases more noticeably as the test parameters move farther from the training range. Nevertheless, the main results in Tables 2 and 3 confirm that the trained hypernetwork still accelerates fine-tuning for these OOD cases.

| OOD shift | Method | RMSE ($10^{-2}$) $\downarrow$ | Rel. Error (%) $\downarrow$ |
|---|---|---|---|
| No shift | HD-direct | $2.7 \pm 1.3$ | $8.9 \pm 4.0$ |
| $\delta = 0.1$ | HD-direct | $10.7 \pm 14.7$ | $51.1 \pm 78.9$ |
| $\delta = 0.2$ | HD-direct | $26.1 \pm 20.4$ | $139.2 \pm 168.4$ |
| $\delta = 0.3$ | HD-direct | $30.8 \pm 24.0$ | $222.8 \pm 295.0$ |
| $\delta = 0.4$ | HD-direct | $34.0 \pm 24.5$ | $391.3 \pm 553.1$ |

Table 5: Direct evaluation of Out-of-distribution on the `Burgers1D-simple` dataset with varying region shift $\delta$.

| Diameter | Method | RMSE ($10^{-2}$) $\downarrow$ | Rel. Error (%) $\downarrow$ |
|---|---|---|---|
| $l \approx 1.20$ | HD-direct | $18.8 \pm 19.1$ | $20.5 \pm 20.8$ |
| $l \approx 1.25$ | HD-direct | $20.5 \pm 27.0$ | $21.7 \pm 28.5$ |
| $l \approx 1.30$ | HD-direct | $19.9 \pm 12.9$ | $20.5 \pm 13.3$ |
| $l \approx 1.35$ | HD-direct | $37.0 \pm 16.9$ | $37.2 \pm 16.9$ |
| $l \approx 1.40$ | HD-direct | $51.2 \pm 19.2$ | $50.1 \pm 18.8$ |

Table 6: Direct evaluation of Out-of-distribution generalization on the `Wave2D` dataset with varying domain diameters $l$.

# I ERROR PLOTS

## I.1 BURGERS1D-COMPLEX.

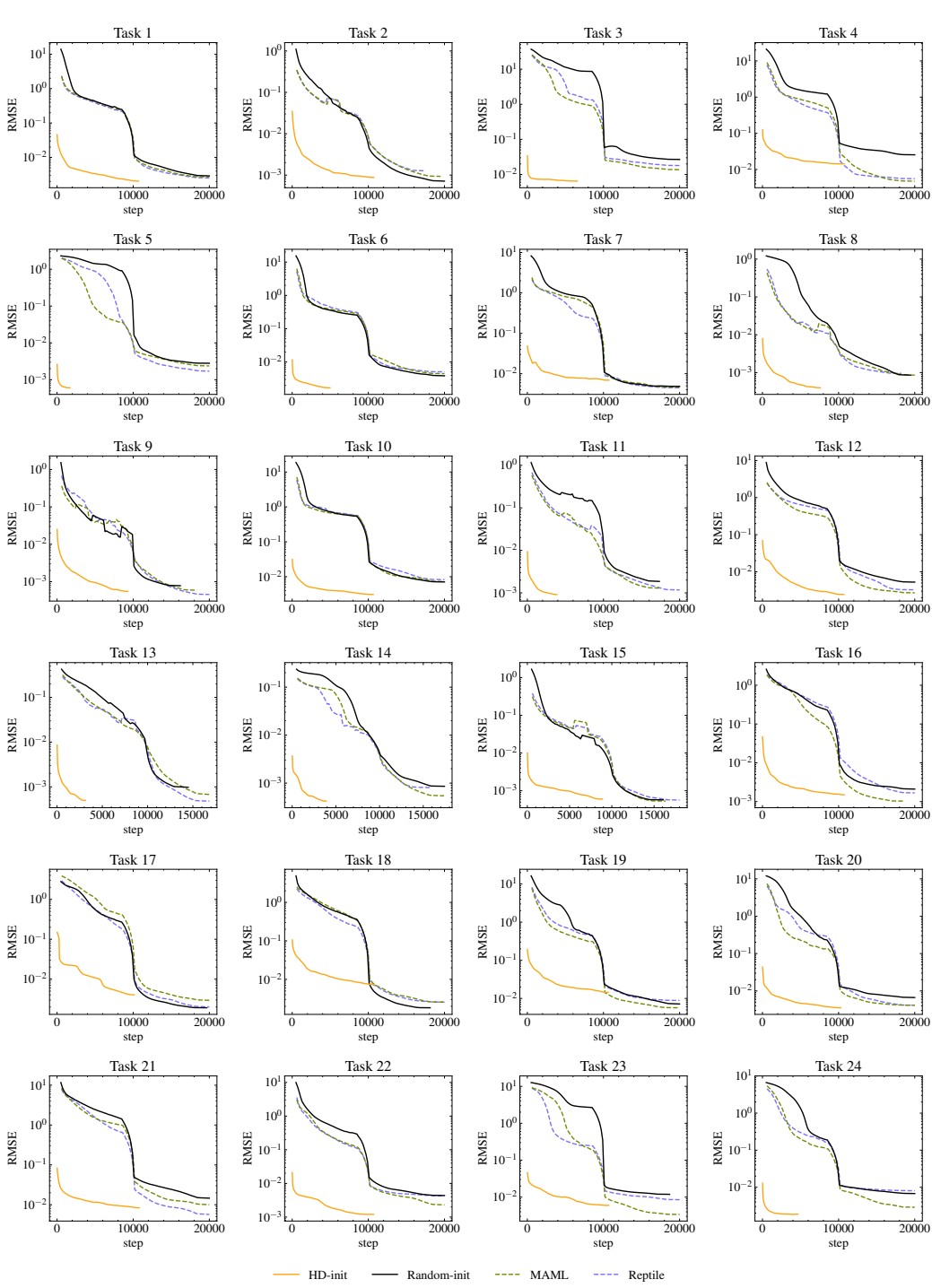

Figure 5: 24 examples of error curves for the `Burgers1D-complex` dataset. The step count denotes training iterations until stopping criteria. In several cases, under the same training budget, PINNs with random initialization fail to reach similar accuracies. In contrast, HD-init provides more favorable starting points, accelerates training and achieves comparable or better accuracies with fewer iterations.

## I.2   WAVE2D.

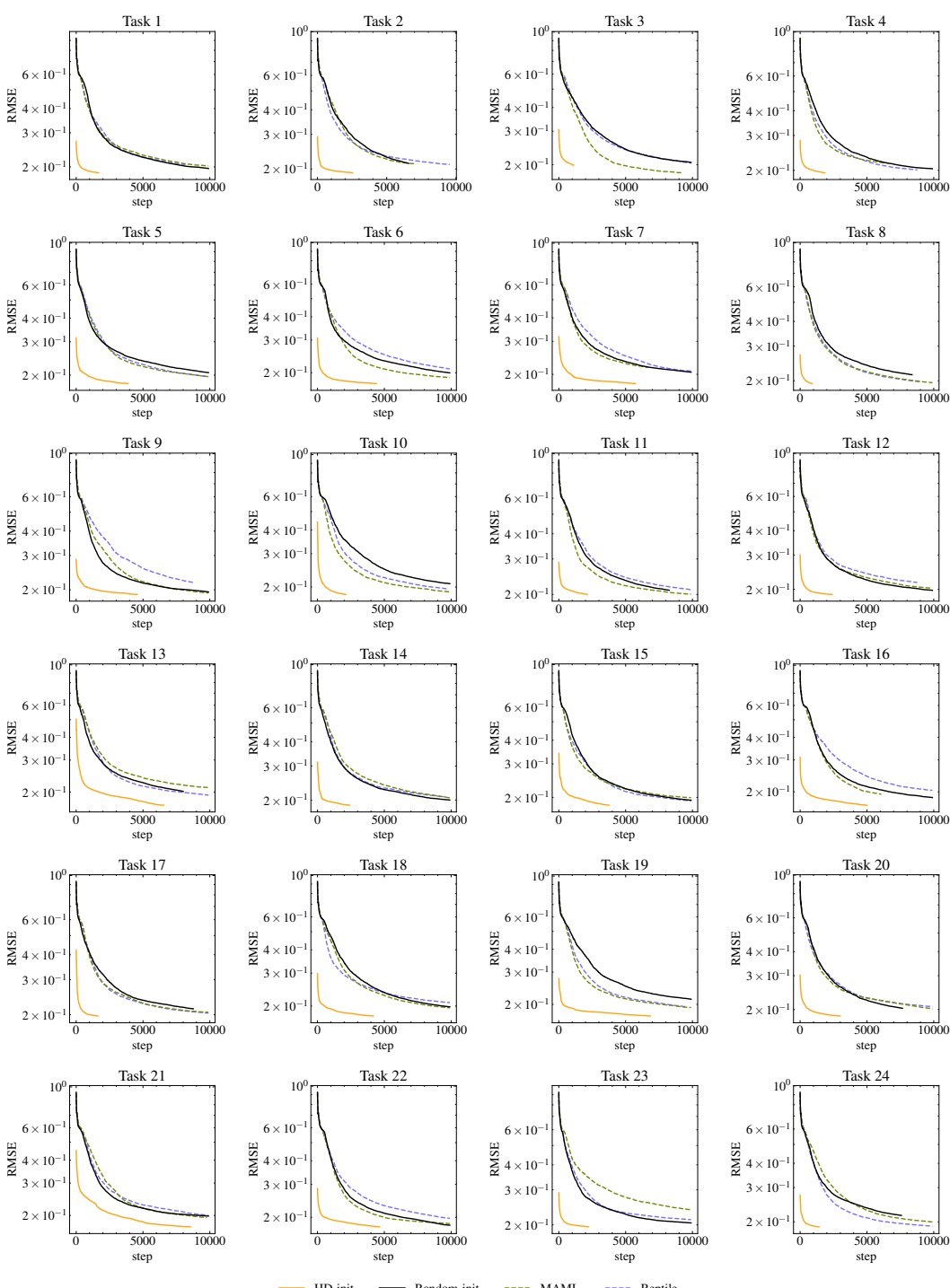

Figure 6: 24 examples of error curves for the Wave2D dataset. The step count denotes training iterations until stopping criteria. In several cases, under the same training budget, PINNs with random initialization fail to reach similar accuracies. In contrast, HD-init provides more favorable starting points, accelerates training and achieves comparable or better accuracies with fewer iterations.

# J    PDE TRAJECTORIES

## J.1    BURGERS1D-SIMPLE.

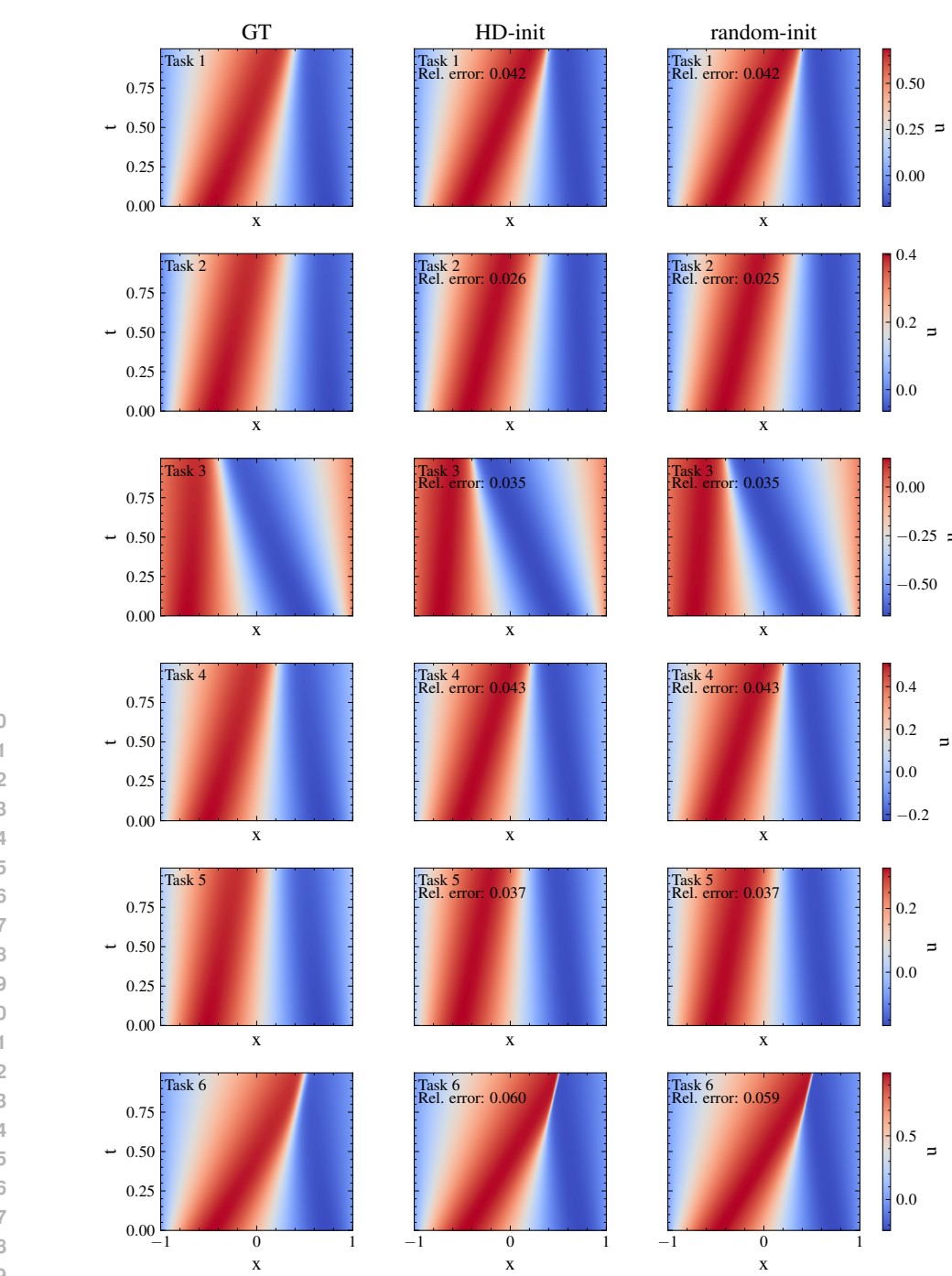

Figure 7: 6 examples of predicted trajectories for the Burgers1D-simple dataset.

## J.2 BURGERS1D-COMPLEX.

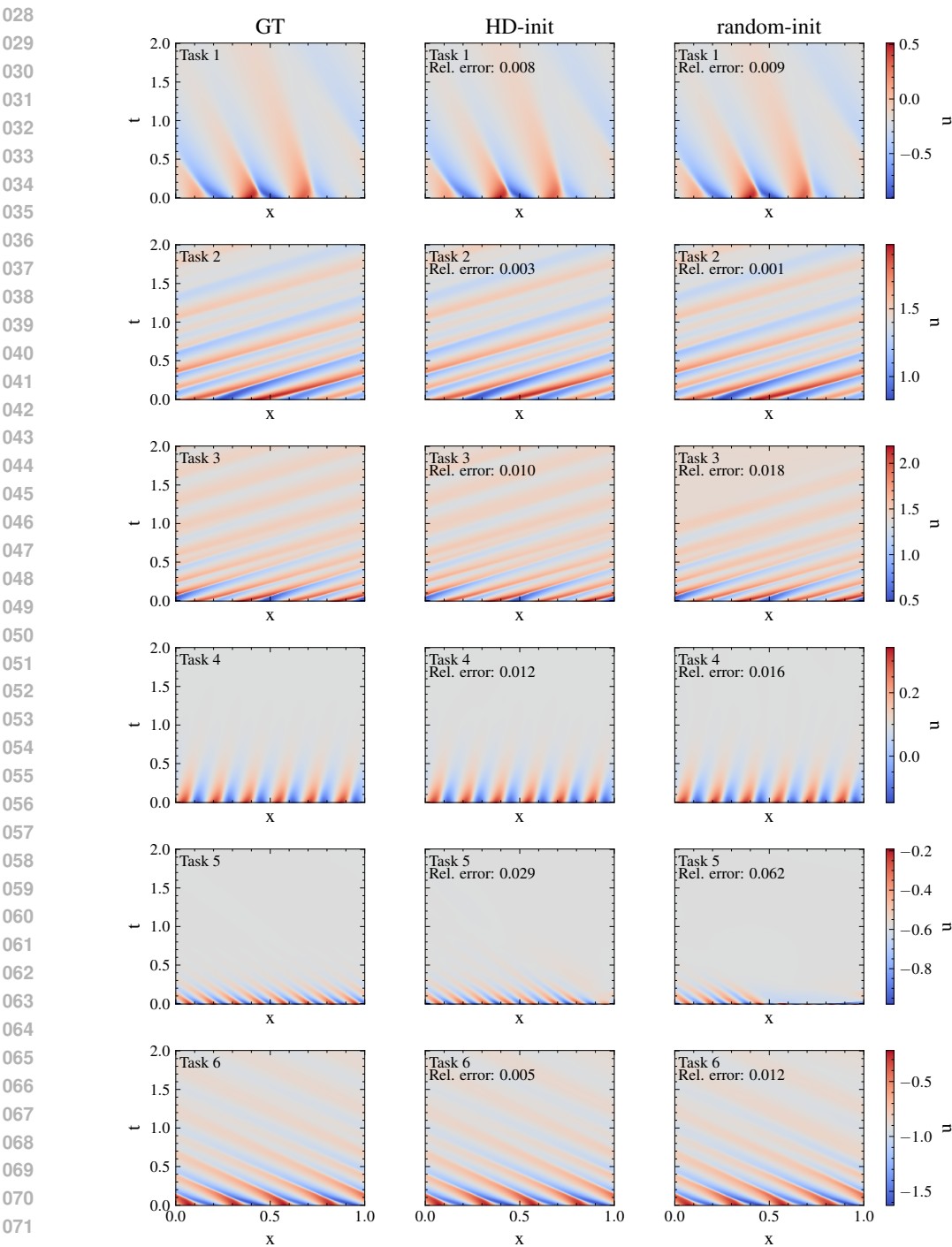

Figure 8: 6 examples of predicted trajectories for the `Burgers1D-complex` dataset.

## J.3  WAVE2D.

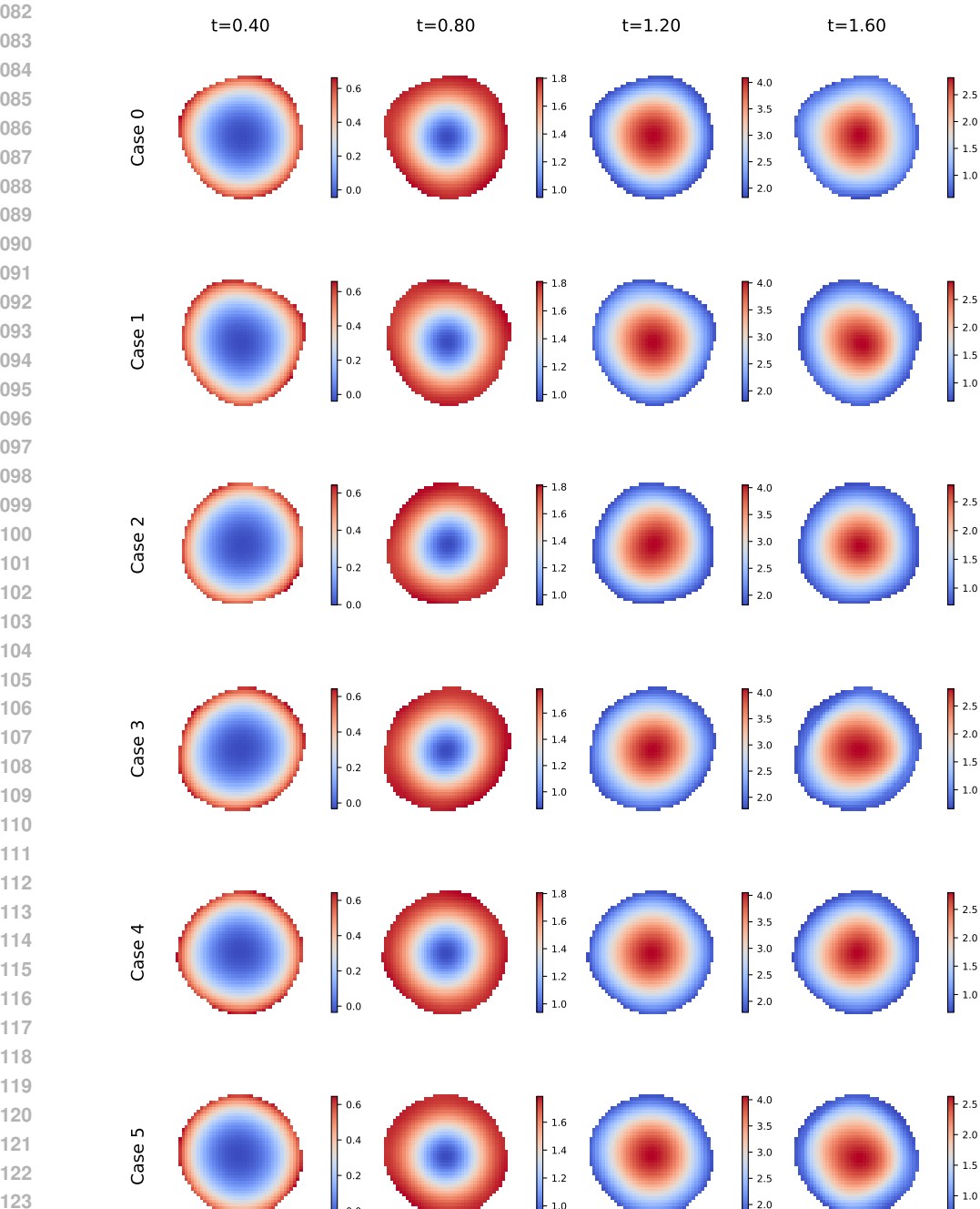

Figure 9: 6 examples of predicted trajectories for the `Wave2D` dataset.

