# OpenReview forum: "Fast Physics-Informed Learning via Diffusion Hypernetworks"
_ICLR.cc/2026/Conference — ICLR 2026 Conference Withdrawn Submission_

### Official Review · Reviewer_KhnP · 2025-10-17

**Soundness:** 2
**Presentation:** 3
**Contribution:** 2
**Rating:** 4
**Confidence:** 4

**Summary:**

This paper proposes a novel diffusion-based hypernetwork that distills knowledge from training data to substantially accelerate PINN training, termed HD-PINN. Specifically, a denoising diffusion probabilistic framework is introduced to generate PINN weights conditioned on PDE parameters. Experiments demonstrate that initializing models with the proposed method effectively enhances training efficiency across multiple tasks. Furthermore, out-of-distribution (OOD) experiments verify the potential of the proposed approach.

**Strengths:**

- The authors have provided code for review, which enhances the reproducibility of this work.
- This paper is easy to follow.
- This paper presents a promising framework.

**Weaknesses:**

- The paper lacks a sufficiently clear description for introducing a diffusion-driven hypernetwork:
    - The reason for introducing hypernetwork: Using PDE/IC/geometric parameters as inputs to PINNs has already been shown to achieve strong generalization after training. This approach has proven highly effective in sufficiently complex tasks (e.g., Parameterized 3D Heat Sink, NVIDIA SimNet™: An AI-accelerated multi-physics simulation framework).
    - The reason for introducing diffusion-based methods: Directly predicting PINN weights via a hypernetwork (without diffusion) is also feasible, more straightforward, and commonly employed.
- Due to the above weakness, the experiments are insufficient. The authors should conduct thorough comparisons with parameterized PINNs (e.g., as in "Parameterized Physics-informed Neural Networks for Parameterized PDEs") to demonstrate the advantages of hypernetwork-based methods in generalization and OOD problems.
- The paper lacks comprehensive ablation studies to evaluate the effectiveness of components in HD-PINN.
- Although the paper presents a promising framework, the model lacks novelty, and the analysis is not sufficiently in-depth.

**Questions:**

Please refer to the weaknesses.

---

### Official Review · Reviewer_Kvqt · 2025-10-27

**Soundness:** 2
**Presentation:** 3
**Contribution:** 2
**Rating:** 2
**Confidence:** 4

**Summary:**

The paper proposes HD-PINN, a diffusion-based hypernetwork that generates or initializes the weights of physics-informed neural networks (PINNs) conditioned on PDE parameters. The authors claim it can drastically reduce training time for parametric PDE families.

**Strengths:**

Conceptual novelty: Using diffusion models to generate PINN weights is an interesting and original idea.

Reasonable speed-ups: On simple testbeds, the initialization reduces training time by about 46–60%.

Clean formulation: The method is well-explained, and code is provided for reproducibility.

**Weaknesses:**

Limited scope of experiments: All tests are on low-dimensional, well-behaved PDEs: 1D Burgers’ and 2D Wave equations. These are relatively smooth, low-chaotic systems where even training from random initialization converges easily (see Table 1 — only seconds for Burgers1D). The method’s claimed acceleration is thus on problems where the baseline is already trivial.

No evaluation on difficult or realistic PDEs: There is no test on stiff, chaotic, or discontinuous systems (e.g., Navier–Stokes turbulence, shock-dominated Burgers, Allen–Cahn, Kuramoto, Sivashinsky, or reaction–diffusion). Without such tests, it’s unclear whether the diffusion-generated weights encode meaningful inductive bias beyond simple regression.

Questionable generalization: The out-of-distribution (OOD) tests shift only scalar parameters (e.g., alpha range or domain diameter), and even small shifts lead to noticeable degradation. There is no validation under truly different boundary conditions or nonlinear parameter regimes.

Comparison with Neural Operators: The HD-PINN can solve a class of PDEs with its diffusion. However, neural operators can also solve a class of PDEs with a certain parametrization. I am unsure what the benefits are compared with neural operators. And such a comparison is lacking. I.e., No comparison with modern operator-learning methods (FNO, DeepONet) that already generalize across PDE families more effectively.

Scalability concerns: The hypernetwork is extremely large (12-layer DiT with >1k hidden dimension) relative to the tiny PINNs (3–4 layers). The paper does not quantify the training cost of the hypernetwork itself, which may exceed the total savings. Also, what is the cost comparison with respect to neural operators that do not require additional networks?

**Questions:**

See weakness.

While the idea of diffusion hypernetworks for PINNs is conceptually interesting, the current experimental evidence is insufficient.
The method is only validated on simple, low-dimensional PDEs where training is already easy. It does not demonstrate robustness to chaotic dynamics, discontinuous solutions, or high-dimensional physical systems — the true bottlenecks for PINNs.

---

### Official Review · Reviewer_DkTU · 2025-11-01

**Soundness:** 3
**Presentation:** 3
**Contribution:** 3
**Rating:** 4
**Confidence:** 4

**Summary:**

This paper proposes a diffusion hypernetwork framework for PINN training. The diffusion models condition on PDE parameters (e.g., initial conditions). Once trained, the hypernetwork can either directly produce PINN weights for simple PDEs or provide good initializations. It is tested on three PDE problems and compared with other meta learning baselines.

**Strengths:**

- The integration of diffusion hypernetwork and PINN training is novel.
- The method also shows some robustness to out-of-distribution PDE parameters

**Weaknesses:**

- Costs and Efficiency. The paper presents training time reductions as a major contribution but does not account for the cost of generating the training dataset and training a DiT model, which are significant. Additionally, the inference cost of the diffusion sampling process is not reported. These make the practical efficiency gains unclear.
- Limited Evaluation. The experiments only cover two types of PDEs (1D Burgers and 2D wave) on low-resolution domains. This raises serious questions about whether the approach can scale to challenging PINN problems.
- Missing Baselines. The paper mainly compares against random initialization and two meta-learning methods (MAML and Reptile), but there are tons of PINN variants that significantly boost accuracy and training time. These methods can be found in any literature review about PINNs.

**Questions:**

- Why use a hypernetwork instead of directly generating solutions (for example, DiffusionPDE and FunDPS)? This architectural choice needs justification.
- A comparison with simple transfer learning (training a PINN on an average or another problem, then fine-tuning) would be valuable.
- The inference stage is not explored: How many steps are needed? How sensitive is it? What's the trade-off between inference steps and final accuracy?

---

### Official Review · Reviewer_Z15p · 2025-11-01

**Soundness:** 2
**Presentation:** 2
**Contribution:** 2
**Rating:** 2
**Confidence:** 4

**Summary:**

In this paper, the authors use diffusion models conditioned on the coefficients of a PDE to provide better initialization, thereby improving training time and performance. They further test their approach on three cases: two involving the 1D Burgers equation and one involving the 2D Wave equation.

**Strengths:**

- The proposed initialization significantly accelerates training, potentially enabling few-shot training for PINNs.

**Weaknesses:**

- While the paper compares training time and accuracy, it is unclear whether the models are useful in scenarios where traditional computational fluid dynamics (CFD) would struggle. The experimental cases presented do not demonstrate such challenging situations.
- Although training time improves for the 2D Wave equation, it is not evident whether out-of-distribution (OOD) performance is substantially better than random initialization. The same concern applies to the Burgers cases.

**Questions:**

- Are any of the experimental cases particularly difficult to solve? If so, what makes them challenging?
- Could the authors provide experiments on a case that is challenging for CFD methods?

---

### Note · Authors · 2025-11-14

I have read and agree with the venue's withdrawal policy on behalf of myself and my co-authors.